# Evaluation framework study assessing the role, applicability and adherence to good practice of planning support tools for allocation of development aid for health in low-income and middle-income countries

Itamar Megiddo ,[1] Shona Blair,[1] Davood Sabei,[1] Francis Ruiz ,[2,3] Alexander D Morton [1]

[1]Department of Management Science, University of Strathclyde, Glasgow, UK
[2]Department of Global Health and Development, London School of Hygiene & Tropical Medicine, London, UK
[3]Center for Global Development, London, UK

**Correspondence to**
Dr Itamar Megiddo;
itamar.megiddo@strath.ac.uk

## ABSTRACT

**Objectives** Allocation of development aid for health is controversial and challenging. In recent years, several planning-software tools have promised to help decision-makers align resource allocation with their objectives, more clearly connect prioritisation to evidence and local circumstances, and increase transparency and comparability. We aim to explore these tools to provide insight into their fitness for purpose and suggest future directions to fulfil that promise.

**Design** We identified seven tools that met the inclusion criteria and developed an evaluation framework to compare them along two dimensions for assessing fitness for purpose: ability to produce analyses adhering to principles laid out in the International Decisions Support Initiative (iDSI) Reference Case for health economic evaluations; and resources required, including expertise and time. We extracted information from documentation and tool use and sent this information to tool developers for confirmation.

**Results** We categorise the tools into evidence-generating ones, evidence-syntheses ones and process support ones. Tools' fitness for purpose varies by the context, technical capacity and time limitation. The tools adhere to several reference case principles but often not to all of them. The source and underlying assumptions of prepopulated data are often opaque. Comparing vertical interventions across diseases and health system strengthening ones remains challenging.

**Conclusions** The plethora of tools that aid priority setting in different ways is encouraging. Developers and users should place further emphasis on their ability to produce analyses that adhere to prioritisation principles. Opportunities for further development include using evidence-generating tools and multicriteria decision analysis approaches complimentarily. However, maintaining tool simplicity should also be considered to allow wider access.

## INTRODUCTION

National health planning tools that simplify the process of priority setting have the

## STRENGTHS AND LIMITATIONS OF THIS STUDY

⇒ The study presents a framework for evaluating tools for national health planning, while noting that context matters.
⇒ The tables provided in the study allow for quick comparison of assumptions, features and required resources across the tools, providing information on their fitness for purpose in a given context.
⇒ The study was limited to desk-based research, which included extracting information from guidance document and peer-reviewed publications, as well as conducting rudimentary analyses using the tools.
⇒ By focusing on publicly available information, the study team avoided having more information than country-level analysts or other stakeholders who use the tools' outputs.
⇒ In-depth interviews with stakeholders of prioritisation could provide additional insight into the challenges of tool development and their future direction.

potential to aid in adhering to the guiding principles of prioritisation. Allocation of development aid for health (DAH) is always controversial and challenging, and the emergence of COVID-19 as a fourth major internationally endemic communicable disease exacerbates the difficulty of overcoming these challenges in a more constrained financial environment. The guiding principles suggest that prioritisation should transparently reflect the values of the country—represented through stakeholders—and this requires a shared understanding across stakeholders of the prioritisation process and the evidence used in this process.[1] Gaining this shared understanding may be difficult due to the differing knowledge base among

stakeholders on health interventions and technical methods for evaluating them. In recent years, research groups and global health and development organisations have developed tools for national health planning in low-income and middle-income countries (LMICs). These tools simplify comparison across programmes and can inform the discussion in the prioritisation process and help develop an evidence base that supports decision-makers at the country level.

It is important to realise that tools support rather than make decisions. A key insight from health technology assessment experience in high-income countries is that effective priority setting requires both technical methods and tools on the one hand and institutions and institutional processes on the other.[2] A number of different process frameworks have been advocated for prioritisation decisions: Socio-Technical Allocation of Resources (STAR),[3] Programme Budgeting and Marginal Analysis,[4] Accountability for Reasonableness[5] and Evidence-Informed Deliberative Process.[6] Though tools and the processes they support are often difficult to separate, in this paper, we are agnostic about how the prioritisation process should be structured. We define a national health planning tool as a software package that requires input and produces outputs specifically designed to support country-level decision-makers in planning. In the rest of the paper, we restrict our analysis and discussion to tools that provide a quantitative analysis, but their purpose may be to generate evidence for use as part of the prioritisation process or to support the process in other ways. Our focus is on these tools' applicability and ability to support or produce analyses that are usable within the context of some structured prioritisation process.

Tool developers make several choices that influence the tools' applicability in these contexts, including ones affecting the required level of expertise to use them and users' ability to adhere to good practice. Tools can offer users more or less flexibility, though they all limit this flexibility to some extent. Unlimited flexibility does away with the tool and leaves all decisions up to the analysts conducting the prioritisation exercise. This flexibility has benefits such as improving the analysts' ability to use the most novel methods and not restricting contextualisation (beyond data availability limitations). However, it also reduces access to conducting prioritisation to those with very high levels of expertise and increases the time costs of a given exercise. The choices made by tool developers limit that flexibility while defining the inherent 'tool opinion' about what ingredients are important for its given purpose. Choices that restrict specific inputs (eg, the number of years into the future considered) may limit users' ability to adhere to certain prioritisation guiding principles, while in other cases, they can guide or steer—and in some cases force—users to adhere to certain principles, exhibiting that 'tool opinion'.

Further, the choices made on the presentation of tools' outputs affect accessibility for users of these outputs. The sheer range of stakeholders that contribute to priority setting can lead to demands for different types of outputs. Their differing knowledge base can contribute to information asymmetries in the prioritisation process and the evidence it uses. The Global Fund to Fight Aids, Tuberculosis and Malaria (GFATM), for example, uses country coordinating mechanisms (CCMs) to submit applications and govern its DAH funds. These CCMs are responsible for empowering and representing the views of stakeholders from across sectors (eg, academic institutions, civil society, government, those with lived experience, non-governmental organizations, and private institutions), and they are asked to submit an application for funding that includes a description of their prioritisation process.[7] A tool with output that simplifies the understanding of evidence or improves the process to incorporate the disparate voices would go a long way toward achieving the vision for CCMs. However, asking tools to cater to all of these stakeholders as well as to donor agendas—that often require information on impact, value for money and equity—is a tall order.

The global health community has a limited grasp on the success of these tools' fulfilment of the promise to increase access to conducting and using the outputs of prioritisation exercises, in a way that is consistent with the principles of good prioritisation. This gap makes it difficult to decide on the context in which one tool should be used rather than another. The World Bank has explored tools for allocative efficiency within the context of HIV, explaining the purpose of a few tools and broadly discussing some of their limitations. The literature on good principles for infectious disease and health economic modelling in LMICs and communication of these models is more mature.[1 8] A recent study, compared COVID-19 models along dimensions of model characteristics and reporting.[9 10] Based on these elements, the study defined a framework for thinking about the models' fitness for purpose, setting questions to guide both output users—policy-makers in this case—and analysts that adapt these models to contextualise an analysis.[9] Both of these audiences are also important for characterising national health planning tools that support priority setting.

Our paper aims to provide an understanding of these tools' characteristics that influence their suitability for specific purposes and to gain insights into their future directions of development and use. We develop an evaluation framework that considers the tools' ability to produce analyses or processes that adhere to good practice and the onus they put on their users in terms of expertise and time. We explore the availability of transparent documentation on the tools and the ease of navigating this documentation. We further describe the types of outputs they produce. We do not rank the tools or suggest that any are inadequate for some part of the priority setting process. In the following sections, we describe our methodology for developing and using our evaluation framework, describe our findings, and discuss conclusions and recommendations.

## METHODS

### Tool selection

We selected planning software tools that aid country-level decision-makers in resource allocation based on the following inclusion criteria: the software package (1) provides a quantitative analysis, (2) is applicable for multiple countries, (3) can be adapted to use new data and (4) is available in the public domain during the evaluation period.

In our search strategy, we sought to identify relevant tools by leveraging the knowledge of our project team and partners and surveying them as well as conducting targeted searches of country investment cases. We followed the search by filtering the tools through the inclusion criteria. Since there are no standard terms that can be used to describe these software tools as a class, we searched for tools that have been previously employed in resource allocation decisions or have received recent investments and backing from global health organisations, indicating their potential future use. The cut-off data for tools we considered was January 2021. Our partners have been active in the field for many years and possess good knowledge of the landscape. Nonetheless, we augmented this knowledge with the targeted searches to not overlook relevant tools.

During the initial step, we conducted discussions with our partners and identified several pertinent tools, including the OneHealth Tool in combination with the Spectrum suite, Optima, Health Interventions Prioritisation (HIP) tool, PriorityVax, Country-led Assessment for Prioritisation on Immunisation (CAPACITI) decision-support tool, STAR and Cascade. In the subsequent step, in targeted searches, we and project partners at the Centre for Global Development reviewed funding requests and investment cases submitted to GFATM and Global Financing Facility, a multistakeholder partnership hosted by the World Bank.[7 11 12] This search confirmed the use of several tools we had identified in the initial step

and additionally identified EQUitable Strategies to Save Lives Tool (EQUIST).

Finally, we identified seven tools that met our criteria among the tools we found in our search. LiST is part of the Spectrum suite, but we include it separately as it has been used standalone for many analyses. We excluded two tools we found in our initial search as they did not meet inclusion criteria (1) and (4); STAR and Cascade, respectively. STAR is a process framework for allocation and does not include a tool that provides quantitative outputs. Cascade is an online tool, and its website was down when the analysis was conducted.

### Development of evaluation framework

The evaluation framework is based on good practices of economic analyses as outlined by the iDSI Reference Case[1] as well as on the requirements for using the tools. The reference case lists eleven principles for economic evaluations (table 1), and these were adapted as relevant for the resource allocation tools and organised in a set of four topics (with corresponding tables): modelling assumptions, model inputs, model outputs, and validity and transparency. Two additional topics (and corresponding tables) that do not correspond to the reference case principles provide an overview of the tools and describe the resources—in time and expertise—required for using the tools and understanding their outputs. Table 1 describes the corresponding principles for each evaluation table.

We examined whether the tools allow users to adhere to these principles, as violating them in a particular prioritisation exercise will also depend on how the tools are used. We dropped principle 2—the comparator accurately reflects the decision problem—which was particularly difficult to capture without considering the tool users' decisions. However, we discuss the limitations to tools' adaptability that may prevent extending analyses to consider additional diseases and interventions (Tools overview and purpose). This discussion should provide

**Table 1** Evaluation framework and the iDSI reference case principles

| Evaluation framework table | Principles |
| --- | --- |
| Tool overview and purpose | Does not correspond to principles |
| Modelling assumptions adherence to reference case | Principle 6—Time horizon and discounting<br>Principle 7—The analysis perspective (non-health effects and costs outside the health budget should be included) |
| Resources and data requirements | Resource requirements do not correspond to principles<br>Principle 3—Systematic and comprehensive evidence gathering<br>Principle 5—Costs reflect all resources used<br>Principle 8—Heterogeneity in population subgroups should be explored |
| Outputs | Principle 4 – Appropriate measures of outcome<br>Principle 8—Heterogeneity in population subgroups should be explored<br>Principle 9—Uncertainty<br>Principle 10—Impact on budget and other constraints identified<br>Principle 11—Explore equity |
| Transparency and validity | Principle 1—Transparency |
| iDSI, International Decisions Support Initiative. | |

sufficient information on the limitations to comparators the user can evaluate.

Each table consists of items used in the data extraction process. Items do not evaluate appropriateness or adherence to the principles. Rather, they report information about the tools characteristics that can be used to evaluate adherence to these principles as well as the level of knowledge needed to use them.

### Data extraction

Two members of the research team extracted information for the seven tools we evaluated on the items identified for each table in the evaluation framework. They experimented with the tools and extracted information based on documentation and publications on the tool (see documentation in online supplemental materials 1). To assess usability, they explored tool documentation and, where available, identified the level of user expertise developers assumed or suggested. They then developed and implemented rudimentary analyses using the tools' interfaces to understand how quickly they could set up an analysis after reading the documentation. After this extraction from documents and experiments with the tools, the entire research team then discussed the data in the evaluation tables to develop the analysis.

### Patient and public involvement

Patients and the public were not involved in this study.

## RESULTS
### Tools overview and purpose

The tools can be categorised along different dimensions that help identify their role in a prioritisation exercise, including by their aims and diseases and interventions they include in an analysis (table 2). Apart from EQUIST and Optima, the tools are not limited by diseases they can analyse.

Further exploring the approaches and methods the tools use, we can categorise them into ones that generate new data (eg, through impact and costing models), ones that synthesise existing data, and ones that focus on problem structuring and process (table 2). Tools that generate new data produce predictions of intervention impact using

| Table 2 | Tools overview and purpose | | |
|---|---|---|---|
| | **Stated aim(s) of tool** | **Approach and theoretical foundations** | **Adaptability to extend (eg, custom interventions)** |
| CAPACITI | Support evaluation of immunisation options incorporating values and programme context | MCDA embedded in broader problem-structuring framework | Focus on immunisation programmes but can extend to other |
| PriorityVax | Support vaccine prioritisation; clarify data uncertainty impact; facilitate discussion; clarify stakeholder views | MCDA embedded in broader problem-structuring framework | Focus on immunisation programmes but can extend to toher |
| HIPtool | Support health intervention prioritisation at country level | Evidence-synthesis | Yes |
| Optima | Analyse and project HIV and TB epidemics; determine optimal resource allocations | Impact model: dynamic compartmental models; Costing: logistic cost function model; Global optimisation to minimise outcome (eg, DALYs) or costs to achieve defined outcomes | No |
| OneHealth (spectrum suite) | Support to inform national healthcare planning and resource needs at country level, strengthening health system analysis, costing and finance | Impact model: module specific (eg, LiST for child and maternal health interventions, or TIME, a dynamic compartmental model for TB); Costing: ingredients-based costing | Custom interventions can be added |
| LiST | Estimate the impact of scaling up MNCH&N interventions in LMICs | Impact model: static equation-based model; costing: ingredient-based costing | Focus on MNCH&N; Custom interventions can be added; only LMICs |
| EQUIST | Estimate the impact of scaling up MNCH&N interventions in LMICs | Bottleneck analysis and statistical analysis of changes to effective coverage; Impact analysis: based on LiST; costing: based on LiST and OneHealth | Focus on MNCH&N; only LMICs |

*Adaptability to extend to new diseases and interventions.
CAPACITI, Country-led Assessment for Priortisation in Immunisation; DALY, Disability adjusted life year; EQUIST, EQUitable Strategies to Save Lives Tool; HIPtool, Health Interventions Priortisation tool; LiST, Lives Saved Tool; LMIC, low-income and middle-income country; MCDA, multicriteria decision analysis; MNCH&N, Maternal, Newborn, Child Health and Nutrition; TB, tuberculosis; TIME, Tuberculosis Impact Model and Estimates.

**Table 3** Modelling assumptions

| | Theoretical basis | Perspective | Time horizon | Forms of uncertainty explicitly considered* |
|---|---|---|---|---|
| CAPACITI | Multicriteria decision analysis (MCDA) embedded in broader problem-structuring framework | Flexible | N/A | Parameter; criteria weighting |
| PriorityVax | MCDA embedded in broader problem-structuring framework | Flexible | N/A | Parameter; criteria weighting |
| HIPtool | Synthesise evidence | Health system | N/A | |
| Optima | Impact model: dynamic compartmental models; costing: logistic cost function model; global optimisation to minimise outcome (eg, DALYs) or costs to achieve defined outcomes | Health system | Do not know | Parameter |
| OneHealth (spectrum suite) | Impact model: module specific (eg, LiST for child and maternal health interventions, or TIME a dynamic compartmental model for TB); costing: ingredients-based costing | Flexible | 100 years fixed time horizon | Parameter |
| LiST | Impact model: static equation-based model; costing: ingredient-based costing | Flexible | Maximum until 2050 | Parameter |
| EQUIST | Bottleneck analysis and statistical analysis of changes to effective coverage based on strategies' bottleneck reduction; impact analysis: based on LiST; costing: based on LiST and OneHealth | Health system | Maximum until 2050 | |

*The two forms of uncertainty in the table include: (1) Parameter—uncertainty in tool input values; (2) criteria weighting—uncertainty in the relative importance of criteria (MCDA specific). Users can explore different scenarios using the tools, including EQUIST, which can be used to assess uncertainty. As this feature is not automated but requires developing new analyses we do not include it in the table. Other forms of uncertainty such as model structural uncertainty are not explicitly considered by the tool and are thus not included.
CAPACITI, Country-led Assessment for Priortisation in Immunisation; EQUIST, EQUitable Strategies to Save Lives Tool; HIPtool, Health Interventions Priortisation tool; LiST, Lives Saved Tool; MCDA, multicriteria decision analysis; N/A, not available.

simulation models. The HIPtool synthesises evidence, relying on evidence generated in external studies. Process tools provide a broader problem-structuring framework such as multicriteria decision analysis (MCDA; eg, PriorityVax). They similarly rely on evidence generated in external studies, but their focus is on the process of the decision-making exercise. Different elements of EQUIST lie in different categories: the tool synthesises evidence to display outputs on bottlenecks and intervention prioritisation; and it guides users on process in a seven-step approach that includes stakeholder discussion.

### Modelling assumptions adherence to reference case
The flexibility the tools provide users (eg, to conduct analyses from different perspectives), their built-in assumptions, and the directions they steer users have implications for users' ability to develop analyses that adhere to the iDSI Reference Case principles (table 3). The principles suggest analyses report a disaggregated societal perspective. Several tools allow users to conduct evaluations from different perspectives. Other tools focus on the health system perspective. A few of the tools restrict the time horizon to 2050, limiting the possibility to consider a lifetime horizon that is recommended by the principles. In evidence-synthesis and process-focused tools, the time horizon depends on the evidence they incorporate and

the transparency and accessibility the tool provides users to its sources (if users use prepopulated data). Most tools allow users to conduct parameter sensitivity and uncertainty analyses. They do not consider stochastic uncertainty—random variation in outcomes—or uncertainty in the analysis structure.

### Resource and data requirement
The required technical expertise and time investment vary with tool complexity (table 1 in online supplemental materials 2). Relatively novice users can implement analyses in tools that merely synthesise information. Though, the users do require an understanding of health economic evaluation terminology. Users need knowledge across a range of modules for more complex tools such as OneHealth and Optima. The tools do not consistently provide guidelines on the time an analysis requires. While tools such as CAPACITI and OneHealth recommend an analysis over 4–6 months, HIPtool does not require significant time investment if analysts and stakeholders use its prepopulated data.

Most tools require intensive data collection to conduct a contextualised analysis, although most, incorporate prepopulated data (table 4). Where they do not provide prepopulated data, they provide guidance on data collection and discuss sources in their literature.[13]

**Table 4** Data requirements

| | Target population | Epidemiological | Intervention coverage | Intervention cost* | Preloaded data |
|---|---|---|---|---|---|
| CAPACITI | Flexible | Flexible | NA | Flexible | No |
| PriorityVax | Flexible | Flexible | NA | Flexible | On request |
| HIPtool | NA | Burden prevalence | Not an input; optimises investment | Intervention unit costs | Yes |
| Optima | By subpopulation (demographic, geography and risk group) and population dynamics† | Several inputs | Over time | Intervention unit costs | Yes (sources in documentation) |
| OneHealth (Spectrum suite) | By subpopulation (sociodemographic and geography)‡; population dynamics† | Several inputs (varies by module) | Over time | Intervention; human-resources, logistics and infrastructure for delivery | Yes (sources in tool) |
| LiST | By subpopulation (sociodemographic and geography); population dynamics† | Incidence, prevalence and health indicators, depending on disease | Over time | Intervention; human-resources, logistics and infrastructure for delivery | Yes (sources in tool) |
| EQUIST | By subpopulations (sociodemographic and geography) | Several inputs | Bottlenecks to effective coverage and their contribution | Yes | Yes (sources in tool); several inputs cannot be updated by the user but can be managed by the 'country administrator' |

Several of the categories are not applicable for evidence synthesis and/or process tools such multicriteria decision analysis (MCDA) ones; flexible—as MCDA tools define criteria as part of the process, these inputs are flexible.

*Users can combine costs to include them under one category; we note how tools disaggregate the cost data further (eg, human resources and logistics).

†The types of data included may vary by module within OneHealth.

‡Population dynamic parameters include ones related to fertility and mortality (exogenous to diseases and interventions evaluated).

CAPACITI, Country-led Assessment for Priortisation in Immunisation; EQUIST, EQUitable Strategies to Save Lives Tool; HIPtool, Health Interventions Priortisation tool; LiST, Lives Saved Tool; NA, not applicable.

The level of guidance varies by tool. Many of the tools reference commonly known global health data sources (eg, the WHO and the Global Burden of Disease (GBD) study). Additionally, they reference specific studies (eg, OPTIMA).[14] We observed differing transparency levels about tools' data sources. PriorityVax, for example, provided a thorough discussion of data sources to support analysis. HIPtool noted its use of the GBD, Disease Control Priorities (DCP3) Essential Universal Health Coverage package and Highest-Priority Package data and 'other secondary sources'. The tool and its documentation do not provide the specific source and underlying assumptions of prepopulated data such as incremental cost-effectiveness ratios (ICERs).

The types of data tools require depends on the tools' aims (table 4). For example, tools that estimate impact require detailed population and epidemiological data. EQUIST, which incorporates a bottleneck analysis, inputs include information about bottlenecks to effective coverage. The MCDA-based tools data requirements depend on the criteria decision-makers and stakeholders deem important. These tools can also include inputs such as ICERs, which tools that generate new data often consider as outputs. HIPtool, similarly, considers ICERs as an input. The HIPtool also requires equity and financial risk protection scores (and provides prepopulated values).

### Outputs

Most tools use visualisation to present outputs on cost, health outcomes (or disease burden) and economic efficiency, with MCDA tools also including aggregated scores (table 5). Both the HIPtool and Optima outputs include a comparison between baseline and optimal scenarios in terms of economic efficiency.

Only some of the tools present uncertainty in their outputs (table 5). Uncertainty bounds are the most common representation of uncertainty among these tools. While the MCDA tools outputs do not explicitly describe uncertainty in results, they recommend users conduct a

**Table 5** Outputs

| | Cost | Health outcomes | Economic efficiency | Type(s) of output | Uncertainty representation | Outputs can be disaggregated into subpopulations |
|---|---|---|---|---|---|---|
| CAPACITI | Yes | Yes | Yes | Aggregate scores, charts, tables | No | No |
| PriorityVax | Yes | Yes | Yes | Aggregate scores, charts, tables | No | Scores based on entire population |
| HIPtool | Yes | Yes | Yes, including optimal package of interventions | Charts | No | No |
| Optima | Yes | Yes | Yes, including optimal package of interventions | Charts, tables | Yes, plots with parameter scenarios | Demographic and geospatial characteristics |
| OneHealth (Spectrum suite) | Yes | Yes | Yes | Charts, tables | Yes, uncertainty bounds | Demographic characteristics |
| LiST | Yes | Yes | Yes | Charts, tables | Yes, uncertainty bounds | Demographic characteristics |
| EQUIST | Yes | Yes | Yes | Charts, tables | No | Demographic charecteristics, wealth and geography |

CAPACITI, Country-led Assessment for Priortisation in Immunisation; EQUIST, Equitable strategies to save lives; HIPtool, Health Interventions Priortisation tool; LiST, Lives Saved Tool.

sensitivity analysis to understand variation due to stakeholder preferences' and criteria values' uncertainty.

Several, but not all, tools allow users to disaggregate results by different demographic dimensions (table 5). While MCDA tools do not present disaggregated outputs, they can include subpopulation outcomes as criteria. EQUIST disaggregates some outputs by wealth. The HIPtool does not produce disaggregated outputs.

### Transparency and validity

The majority of the tools provided some degree of documentation; however, we found variation in the accessibility of the documentation (table 6). All tools have produced

at least one analysis that was published in a peer-reviewed journal. Frequently, such publications were written by the tool developers themselves, although not in all cases. Given the variety of release dates of the tools (table 1 in the online supplemental materials), the lack of independent tool usage does not necessarily imply a lack of tool usage by other groups.

### DISCUSSION AND CONCLUSION

The promise of health planning tools is that they will simplify the process of prioritising health interventions

**Table 6** Transparency and validity

| | Documentation availability* | Limitations explained | Interests of authors declared | Funding declared | Open source | Peer-reviewed publications |
|---|---|---|---|---|---|---|
| CAPACITI | Yes | Yes, but no dedicated limitations section | No | No | Yes | Yes |
| PriorityVax | Yes | Yes, but no dedicated limitations section | No | Yes | No | Yes |
| HIPtool | No (work in progress) | No | No | No | No | Yes |
| Optima | Yes | Yes | Yes | Yes | No | Yes |
| OneHealth (Spectrum suite) | Yes | Yes | Yes | Yes | No | Yes |
| LiST | Yes | Yes | Yes | Yes | No | Yes |
| EQUIST | Yes | Yes, but no dedicated limitations section | No | Yes | No | Yes |

*This included both guidance manuals and technical reports. Other forms of documentation (presentations, online discussions, secondary reports) were available for some tools; however, these have not been included in the evaluation.
CAPACITI, Country-led Assessment for Priortisation in Immunisation; EQUIST, EQUitable Strategies to Save Lives Tool; HIPtool, Health Interventions Priortisation tool; LiST, Lives Saved Tool.

and resource allocation in a manner that consistently adheres to accepted principles while also creating trust and a shared understanding among stakeholders. In this study, we developed and implemented an evaluation framework to describe tool characteristics that convey where the tools succeed and provide insight into potential future directions for tool developers. Our hope is that comparing tool characteristics also helps tool users identify which tool—or tools—are well suited for their prioritisation exercise.

Based on our findings on tool purpose and the types of inputs they require, we group the tools into three non-exclusive categories: evidence-generating, evidence-synthesising and process. Evidence-generating tools produce outputs used in the prioritisation process, including estimates of impact and economic efficiency. They often produce these outcomes using a what-if style analysis that explores potential outcomes if a certain allocation is implemented. In some cases, they explore optimising this allocation, for example, from an economic efficiency perspective. Tools such as LiST and Optima are in this group. Evidence-synthesising tools combine evidence from different sources to provide a comprehensive picture for their users. By evidence-synthesising tools, we mean tools that mainly incorporate data that was generated externally and do not further manipulate this data. The HIPtool would fall in this category. Evidence-synthesising tools inform the discussion in a prioritisation process, and provided the data they synthesise is comparable, they can optimise along the dimension of that data (eg, find the allocation that is most cost-efficient).[15] Lastly, process tools such as the MCDA tools—PriorityVax and CAPACITI—support the process of prioritisation more broadly. They can help capture and balance values which cannot be readily modelled.

Which tools should be used depends on the needs of a specific prioritisation exercise, and the tools can often be complementary rather than competitive. For example, MCDA process tools complement tools which have a more technically developed modelling base for modelling disease dynamics or programme costing (the HIPtool can similarly be used in this manner). Indeed, technical tool outputs can be used as inputs to MCDA tools. MCDA-based tools have the potential to bridge the gap between technical analysis and political decision-making by highlighting trade-offs between dimensions of value that can be modelled and those which are more difficult to model (eg, concerns related to justice or economic spillovers). A welcome development would be the conduct of pilot studies that use both technical modelling tools in tandem with MCDA tools to better understand these synergies.

Technical capacity and time available also influence tool choice. Evidence-synthesis tools that do not generate new data may be the simplest to use, particularly if they include prepopulated data. However, this comes at a cost, as their data may come from settings that are different from the context of the given prioritisation exercise. Evidence-generating tools typically require a higher level of technical expertise. Tool design that restricts users' inputs and guides them through the process of conducting an evidence-generating analysis step-by-step simplifies the job for analysts. These restrictions also come at the cost of flexibility, for example, to contextualise prioritisation exercises. Reducing the amount of data inputs required in these tools also simplifies the process for users, with a similar cost. While MCDA tools may be relatively simple to navigate, MCDA usually requires experienced facilitators alongside the tools to incorporate the views of relevant stakeholders.

No matter the choice of tools, analyses should adhere to good principles of prioritisation, and the tools must provide the opportunity to do so. In our framework, we use the iDSI Reference Case, which focuses on economic evaluations, to explore tools' adherence to good principles.[1] Though prioritisation involves more than economic evaluation, many of the principles are relevant. In our evaluation, we found that some of the tools do not currently allow users the opportunity to adhere to all principles. For example, a number of tools have limits on the time horizon that may reduce consideration of relevant costs and benefits. Though we focus on the iDSI Reference Case, tools that generate evidence on impact should follow modelling principles in their respective areas (eg, the disease areas or type of modelling methodology).

The principle of transparency is in some cases violated when tools prepopulate inputs to simplify the process for users but do not provide information about the sources of this data and their underlying assumptions. Prepopulating tools with data and asking users to edit inputs only where needed (to contextualise inputs) is quite attractive; it significantly simplifies the process of implementing an analysis. However, the data should be transparent in its sources and assumptions. Though many tools explicitly state data sources, not all do so. In some cases, they broadly state sources but do not specify them for specific inputs. Prepopulated data may have underlying assumptions, and these are often not clear to tool users. This lack of transparency can reduce trust by the user and may lead to analyses that combine data in a manner that does not necessarily make sense. Tools that do provide prepopulated data offer the option to change this input, potentially mitigating any lack of transparency. However, the mere existence of the prepopulated data may steer users in one direction.

To avoid inappropriate analyses, tools should provide clear documentation on 'how' they work in addition to how to use them. This recommendation similarly falls under the principle of transparency. Most, but not all, tools already provide sufficient documentation guiding users and publications explaining the methodological foundations of models. In some cases, guidance documentation can be simplified and made more accessible; though, we note that the complexity of documentation naturally increases as the complexity of the tool increases (eg, OneHealth). Explanation of the tool's methodological foundation in peer-reviewed publications can help

experts in countries gain trust for the tools. This trust can then be conveyed to other stakeholders. Repositories of the analyses using the tools may also be helpful. Documentation that describes the tool's ownership and process of development and validation can further build trust.

We identified a number of areas where tools can innovate and incorporate outputs that are important for both donors and policymakers. For example, the tools we considered did not make it easy for stakeholders to assess equity. A simple option is to allow disaggregating outputs by subpopulation, and a number of tools already do this; however, tools that do allow disaggregation do not always allow doing this along all dimensions that are important to stakeholders (eg, wealth). Disaggregating outputs would align with the DCP3 approach of extended cost-effectiveness analysis panels.[16] Alternatively, analysts can combine outputs from several runs of a tool using data specific to a given subpopulation each time. However, this puts the onus on tool users (the analysts). Tools could also take novel approaches and help users develop analyses based on distributional cost-effectiveness analysis framework, though data limitations may make this difficult.[17]

An opportunity also exists for building on tools to develop analyses that can compare interventions for different diseases with health system strengthening ones. The OneHealth tool is the most comprehensive among the tools; it incorporates linked modules—which are prioritisation tools themselves focused on specific elements—to be able to make comparisons across diseases and investments to strengthen health systems. That complexity comes at a cost: increased resources from users as well as increased difficulty to validate the consistency of the linked modules, which have been combined over the past two decades and were originally developed by different groups. EQUIST, which includes neonatal, infant and maternal interventions, takes a completely different approach to health system strengthening. It helps identify bottlenecks and estimates the effect of reducing these bottlenecks on effective coverage. It links to other tools such as LiST and OneHealth to then conduct impact and costing analyses.

Our study is limited as it involved only desk-based research that included extracting information from guidance documents and peer-reviewed publications and exploring rudimentary analyses using the tools. Further discussion with tool developers and experienced users would provide a more complete understanding of the tools. However, this limitation can also be viewed as an advantage, as it allowed us to understand the tools based on what is available in the public domain. We held a workshop with tool developers and users, but further interviews with them and stakeholders of prioritisation would provide further insight into the challenges of tool development and their future direction.

Overall, we are encouraged by the plethora of different types of tools available for prioritising resource allocation. There is no one tool that is optimal in every case, and different categories of tools are helpful for different parts of the prioritisation process and for different audiences. As outlined in this discussion, we do see gaps and areas in which the tools included in our analysis can be improved.

**Acknowledgements** We would like to thank participants of the "Why Good Priority-Setting Tools are More Important for Global Health than Ever Before" workshop, which was held in September 2021. The workshop helped refine the research.

**Contributors** Concept and design: IM, FR and ADM. Acquisition of data: SB and DS. Analysis and interpretation: IM, SB, DS and ADM. Drafting of the manuscript: IM, SB and DS. Critical revision of paper for important intellectual content: IM, SB, DS, FR and ADM. Provision of study materials: DS. Obtaining funding: IM, FR and ADM. Administrative, technical, or logistic support: FR. Supervision: IM and ADM, Gaurantor:IM.

**Funding** This work received funding support from the Bill & Melinda Gates Foundation (OPP1202541) through the International Decision Support Initiative (iDSI).

**Disclaimer** The funders did not have a role in writing the manuscript or the decision to submit for publication.

**Competing interests** All authors received support for this study from Bill & Melinda Gates Foundation through the International Decision Support Initiative. ADM and IM reported receiving grants from National Institute of Health Research/ Department of Health and Care Policy Research programme. ADM additionally reported receiving grants from NIHR Global Health Group programme, Chief Scientist's Office Scottish NHS, European Commission, Cancer Research UK and Pancreatic cancer UK. ADM reported receiving book royalties from Springer. IM additional reported receiving grants from the Scottish Funding Council. ADM reported receiving consulting fees from the Bill & Melinda Gates Foundation via the Thailand Ministry of Public Health and from the WHO Global Malaria Programme. IM reported receiving consulting fees from the International Decision Support Initiative. ADM reported receiving payment or honoraria for lectures from the National University of Singapore. ADM reported participation in the Office of Health Economics board.

**Patient and public involvement** Patients and/or the public were not involved in the design, or conduct, or reporting, or dissemination plans of this research.

**Patient consent for publication** Not applicable.

**Provenance and peer review** Not commissioned; externally peer reviewed.

**Data availability statement** All data relevant to the study are included in the article or uploaded as online supplemental information. All data used in the study are publicly available.

**ORCID iDs**
Itamar Megiddo http://orcid.org/0000-0001-8391-6660
Francis Ruiz http://orcid.org/0000-0001-5183-3959
Alexander D Morton http://orcid.org/0000-0003-3803-8517

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
