## [Reviewer comments · BMJ Open]

ARTICLE DETAILS

TITLE (PROVISIONAL)	An evaluation framework study assessing the role, applicability, and adherence to good practice of planning support tools for allocation of development aid for health in low- and middle-income countries
AUTHORS	Megiddo, Itamar; Blair, Shona; Sabei, Davood; Ruiz, Francis; Morton, Alexander

VERSION 1 – REVIEW

REVIEWER	Leroueil, Pascale University of Michigan William Davidson Institute
REVIEW RETURNED	27-Jan-2023

GENERAL COMMENTS	This manuscript is meant to provide an overview of publicly available tools that can be used to support resource allocation in the health space. The authors describe their approach for identifying the tools, and map the characteristics of the tools against the iDSI Reference case principles. This type of summary is useful for both the target users/beneficiaries of the tools, as well as model developers interested in identifying unmet needs. 1. Provide more clarity on who the target beneficiaries of these models are. Specifically, was the intent to focus on tools that would be helpful for decision makers at the country level, or within organizations that provide development aid for health? My suspicion is that it is the former given the tools that have been identified, but the manuscript seems to be framed around decisions needing to be made about development aid for health and this could refer to allocation decisions by donors. Do the authors mean to say the allocation of development aid by countries within their programs? 2. Add more detail related to the search process for identifying the world of potential tools to investigate. The authors state the inclusion criteria, but what was the approach used to define the entire world of potential tools to evaluate? For example, were there specific search terms that were used in Google? Web of Knowledge? PubMed? Also, please include the cut-off date that was used for defining the entire world of potential tools to evaluate. 3. Revise headings of sections to match what is listed under the Evaluation Framework in Table 1. For the most part there is alignment but they are not exactly the same. I recommend accepting this manuscript with the minor revisions mentioned above. Pascale Leroueil, PhD MBA
--

REVIEWER	Kirabo-Nagemi , Charity
-----------------	-------------------------

	The University of Queensland
REVIEW RETURNED	20-Feb-2023

GENERAL COMMENTS	The paper contributes to allocation decisions in external financing for healthcare, which is a topic of great interest and relevance amidst competing priorities, population increase and the increasing interest in value-for-money investments. Overall, this is a well written paper which I recommend for publication, after minor revisions. The following areas should be addressed Key messages; How this study might affect research, practice or policy Lines 29 and 33 in the should be punctuated better to avoid confusion. Introduction Page 7 of 30, line 10. An apostrophe on the word tools. Page 7 of 30, lines 24 -26. The sentence should end at “some extent” such that a new one begins at “Unlimited”. Page 8 of 30 line 35. A comma after the word exercises would help to break up the long sentence. Page 9 of 30, line 6 to 8 should be re-phrased to eliminate the redundancy of “these tools”. Methods Page 11 of 30, line 23. “the limitations tools adaptability to extensions that consider additional diseases” should either be rephrased or punctuated better to make it easier for the reader to understand Page 11 of 30, line 55. “They then developed implemented” shouldn’t this be “developed and implemented”?
---

VERSION 1 – AUTHOR RESPONSE

Reviewer: 1

This manuscript is meant to provide an overview of publicly available tools that can be used to support resource allocation in the health space. The authors describe their approach for identifying the tools, and map the characteristics of the tools against the iDSI Reference case principles. This type of summary is useful for both the target users/beneficiaries of the tools, as well as model developers interested in identifying unmet needs.

- (1) Provide more clarity on who the target beneficiaries of these models are. Specifically, was the intent to focus on tools that would be helpful for decision makers at the country level, or within organizations that provide development aid for health? My suspicion is that it is the former given the tools that have been identified, but the manuscript seems to be framed around decisions needing to be made about development aid for health and this could refer to allocation decisions by donors. Do the authors mean to say the allocation of development aid by countries within their programs?

Response: You are correct that we focus on the tools helpfulness at the country level.

We added the bold text in the following sentence to clarify (Pg 4, end of 1st paragraph):
*These tools simplify comparison across programmes and can inform the discussion in the prioritisation process and help develop an evidence base **that supports decision-makers at the country level.***

We also amended the following sentence (Pg 5, middle of paragraph starting on pg 4):

*We define a national health planning tool as a software package that requires input and produces outputs specifically designed to support **country-level decision-makers** in planning.*

Lastly, we also added to the Methods, Tools selection section (Pg 7, beginning of 2nd paragraph):
*We selected planning software tools that aid **country-level decision-makers** in resource allocation based on the following inclusion criteria: the software package...*

- (2) Add more detail related to the search process for identifying the world of potential tools to investigate. The authors state the inclusion criteria, but what was the approach used to define the entire world of potential tools to evaluate? For example, were there specific search terms that were used in Google? Web of Knowledge? PubMed? Also, please include the cut-off date that was used for defining the entire world of potential tools to evaluate.

Response: We added text to describe our search strategy and relevant dates in more detail (Pg 7 last paragraph – Pg 8 first paragraph; new text in bold):

In our search strategy, we sought to identify relevant tools by leveraging the knowledge of our project team and partners and surveying them as well as conducting targeted searches of country investment cases. We followed the search by filtering the tools through the inclusion criteria. Since there are no standard terms that can be used to describe these software tools as a class, we searched for tools that have been previously employed in resource allocation decisions or have received recent investments and backing from global health organisations, indicating their potential future use. The cut-off data for tools we considered was January 2021. Our partners have been active in the field for many years and possess good knowledge of the landscape. Nonetheless, we augmented this knowledge with the targeted searches to not overlook relevant tools.

During the initial step, we conducted discussions with our partners and identified several pertinent tools, including the OneHealth Tool in combination with the Spectrum suite, the Lives Saved Tool (LiST), Optima, Health Interventions Prioritisation (HIP) tool, PriorityVax, Country-led Assessment for Prioritisation on Immunization (CAPACITI) decision-support tool, STAR, and Cascade. In the subsequent step, in targeted searches, we and project partners at the Centre for Global Development reviewed funding requests and investment cases submitted to GFATM and Global Financing Facility, a multi-stakeholder partnership hosted by the World Bank.^{7,11,12} This search confirmed the use of several tools we had identified in the initial step and additionally identified EQUitable Strategies to Save Lives Tool (EQUIST).

Finally, we identified seven tools that met our criteria among the tools we found in our search. ...

- (3) Revise headings of sections to match what is listed under the Evaluation Framework in Table 1. For the most part there is alignment but they are not exactly the same.

Response: We amended the headings and the Table 1 to align.

Reviewer: 2

The paper contributes to allocation decisions in external financing for healthcare, which is a topic of great interest and relevance amidst competing priorities, population increase and the increasing interest in value-for-money investments. Overall, this is a well written paper which I recommend for publication, after minor revisions.

Response:

The following areas should be addressed

Key messages

- (1) How this study might affect research, practice or policy

Response: This section was removed and replaced by a Strengths and limitations section to adhere to the journal format.

(2) Lines 29 and 33 in the should be punctuated better to avoid confusion.

Response: This section was removed and replaced by a Strengths and limitations section to adhere to the journal format.

Introduction

(3) Page 7 of 30, line 10. An apostrophe on the word tools.

Response: Thank you. Fixed.

(4) Page 7 of 30, lines 24 -26. The sentence should end at “some extent” such that a new one begins at “Unlimited”.

Response: Fixed.

(5) Page 8 of 30 line 35. A comma after the word exercises would help to break up the long sentence.

Response: Added.

(6) Page 9 of 30, line 6 to 8 should be re-phrased to eliminate the redundancy of “these tools”.

Response: The sentence was rewritten as follows (Pg 7, paragraph 1; amendments in bold)
*Our paper aims to provide an understanding of these tools’ characteristics that influence their suitability for **specific purposes** and **to gain insights** into **their** future directions **of development and use**.*

Methods

(7) Page 11 of 30, line 23. “the limitations tools adaptability to extensions that consider additional diseases” should either be rephrased or punctuated better to make it easier for the reader to understand

Response: We rephrased the sentence as follows (Pg 10, paragraph 1)
However, we discuss the limitations to tools’ adaptability that may prevent extending analyses to consider additional diseases and interventions (Tools overview and purpose). This discussion should provide sufficient information on the limitations to comparators the user can evaluate.

(8) Page 11 of 30, line 55. “They then developed implemented” shouldn’t this be “developed and implemented”?

Response: Yes, thank you. Fixed.

VERSION 2 – REVIEW

REVIEWER	Kirabo-Nagemi , Charity The University of Queensland
REVIEW RETURNED	23-May-2023
GENERAL COMMENTS	You have addressed all the comments and made the appropriate revisions. Congratulations on a great paper.